**Subject Category:**
Biology (whole organism)

behaviour/psychology

social learning, associative learning, animal behaviour, animal cognition, animal culture

**Author for correspondence:**
Johan Lind
e-mail: johan.lind@zoologi.su.se

# Social learning through associative processes: a computational theory

Johan Lind[1], Stefano Ghirlanda[2] and Magnus Enquist[3]

[1]Centre for the Study of Cultural Evolution and Department of Zoology, Stockholm University, Stockholm, Sweden
[2]Department of Psychology, Brooklyn College of CUNY, Brooklyn, NY, USA
[3]Department of Zoology, Stockholm University, Stockholm, Sweden

JL, 0000-0002-4159-6926; SG, 0000-0002-7270-9612

Social transmission of information is a key phenomenon in the evolution of behaviour and in the establishment of traditions and culture. The diversity of social learning phenomena has engendered a diverse terminology and numerous ideas about underlying learning mechanisms, at the same time that some researchers have called for a unitary analysis of social learning in terms of associative processes. Leveraging previous attempts and a recent computational formulation of associative learning, we analyse the following learning scenarios in some generality: learning responses to social stimuli, including learning to imitate; learning responses to non-social stimuli; learning sequences of actions; learning to avoid danger. We conceptualize social learning as situations in which stimuli that arise from other individuals have an important role in learning. This role is supported by genetic predispositions that either cause responses to social stimuli or enable social stimuli to reinforce specific responses. Simulations were performed using a new learning simulator program. The simulator is publicly available and can be used for further theoretical investigations and to guide empirical research of learning and behaviour. Our explorations show that, when guided by genetic predispositions, associative processes can give rise to a wide variety of social learning phenomena, such as stimulus and local enhancement, contextual imitation and simple production imitation, observational conditioning, and social and response facilitation. In addition, we clarify how associative mechanisms can result in transfer of information and behaviour from experienced to naive individuals.

## 1. Introduction

Social learning—learning from others—is a key phenomenon in the evolution of behaviour and in the origin of traditions and culture. At the individual level, social learning helps naive

individuals acquire information from more experienced individuals, resulting in behaviours that have positive outcomes or result in avoidance of negative ones. This saves time and energy and reduces individual risk, ultimately enhancing survival and reproduction (e.g. [1–3]). At the group level, social learning enables the transmission of behaviour between individuals and across generations, thus providing the opportunity for the establishment of traditions (*sensu* [4]) and other cultural phenomena [5–12].

Through decades of research, a wide array of social learning phenomena have been described in many animal groups (e.g. [7,11,13–15]), and controversy has ensued regarding both the social learning abilities of non-human animals (hereafter: animals) and the learning mechanisms that underlie the phenomena. At one extreme, social learning in animals is claimed to require no special capacity and to be accounted for by associative learning (e.g. [16–19]). At the other extreme, animal social learning is considered cognitively advanced, almost human-like (e.g. [20,21]). In addition, social learning phenomena are described using a dauntingly diverse terminology, such as stimulus and local enhancement, emulation, social facilitation and imitation. Some regard these terms as identifying different learning mechanisms (e.g. [11,12,22–24]), while others consider them as merely descriptive, with the phenomena themselves amenable to explanation through just a few associative mechanisms [13,25–27]. Setting controversy aside, the diverse social learning terminology reflects a genuine diversity of possibilities for social learning. For example, an animal may simply notice the presence of others, it may observe the behaviour of others or outcomes of behaviours, and it may observe what objects others are interacting with. In addition, an animal can use these observations in many ways, such as to approach or avoid objects and other animals, or to try out other behaviour. Lastly, social information can be available in many contexts, such as when encountering novel objects, when foraging, or when learning about predators.

This diversity notwithstanding, the adaptive value underlying social learning is that inexperienced individuals can learn to behave more efficiently by using social information from experienced individuals. This information transfer is the main focus of our paper, in particular, when it results in naive individuals behaving similarly to experienced individuals. More specifically, we explore the extent to which the social transmission of productive behaviour can be supported by associative learning processes. We first discuss the nature of social information, and then we introduce our model, in which associative learning is guided by genetic predispositions [28]. This model, closely related to optimization algorithms in machine learning that are commonly used in, for example, 'deep learning' studies [29], can learn optimal behaviour in ecologically relevant circumstances. Furthermore, the model has previously been shown capable of producing 'intelligent' behaviour, such as tool use, self-control and planning [28,30]. Drawing inspiration from Cecilia Heyes' extensive work (e.g. [25,27,31]), we ask whether the model can be a satisfactory account of social learning phenomena in animals. We will not analyse in detail specific datasets, but rather investigate whether the model can reproduce the kind of phenomena that are usually included in social learning taxonomies, using as a benchmark the taxonomy by Hoppitt & Laland [11]. We will conclude that associative learning can account for a surprising array of social learning phenomena. In the Discussion, we consider how our model relates to other analyses of social learning.

# 2. Material and methods

There is a rich terminology for the roles of the individuals involved in social learning. The recipient of social information may be called the observer, the young or the naive individual. The provider of social information may be called the model, the demonstrator, the teacher or the experienced individual. In this paper, we use the terms 'learner' for the first individual and 'experienced' animal for the second. With these terms, we intend to emphasize the possibility of information transfer from one individual to the other, without implying that such transfer is deliberate, nor invariably adaptive.

## 2.1. The nature of social observations and the correspondence problem

A simple definition of social learning is that an individual can learn based on stimuli that originate, at least in part, from other individuals [25,27]. We refer to these as 'social stimuli' and treat them like any other stimulus. For example, one animal may observe another eating a fruit, and thereby learn that the fruit is worth eating. In this case, the other animal and its actions provide social stimuli, while the fruit is a non-social stimulus. Typically, observers perceive stimulus compounds that contain both social and non-social elements. We can write such a compound stimulus simply by listing its components

$$S = S_{\text{social}}S_xS_y\ldots,$$

where $S$ is the complete stimulus, $S_{\text{social}}$ is a social element of interest and $S_x S_y \ldots$ represent other stimulus elements, social or non-social. In the previous example, $S_{\text{social}}$ would represent stimuli originating from the observed animal, $S_x$ from the fruit it was eating and $S_y \ldots$ from other parts of the environment. Most social learning situations involve observing a sequence of stimuli with social elements. Consider, for example, an individual that performs behaviour B towards stimulus $S$, resulting in a second stimulus $S'$. We write such a sequence as

$$S \to B \to S', \tag{2.1}$$

where we have used the convention that stimuli are in italic and behaviours in roman type. An example sequence is

$$Fruit \to Eat \to Sweet.$$

When such a sequence is observed by another individual, the stimuli are typically not the same, but they are related. We use brackets to indicate an observer's perception, rather than the perceptions and actions of the individual that performs the sequence. For example, [Eat] refers to the perception of another individual eating. Thus, when perceived by another individual, equation (2.1) becomes

$$S_{\text{social}}[S] \to S_{\text{social}}[B] \to S_{\text{social}}[S'], \tag{2.2}$$

where $S_{\text{social}}$ refers to observer's perception of the other individual. In equation (2.2), we have indicated the presence of social stimuli with $S_{\text{social}}$, but in many situations, it is important to distinguish between different social stimuli, such as a particular display indicating danger rather than the simple presence of another individual. The complexity of social observations will increase further if a sequence of behaviour is considered rather than a single response.

Note that stimuli and actions may be more or less similar from the point of view of the observer and observed individual. Thus, while [Fruit] can bear strong similarity to Fruit, [Sweet] and Sweet will not resemble each other at all: Sweet is a gustatory sensation, while [Sweet] indicates the consequences of this sensation that are visible to others, if any. Likewise, the stimuli experienced while performing the behaviour Eat are altogether different from [Eat], which refers to the perception of others eating. In other words, to be effective, the social learner must somehow bridge the gap between its own perspective and that of other individuals [32]. This is referred to as the 'correspondence problem' in the social learning literature [33,34]. As hinted above, the correspondence problem applies to varying extent. The stimulus situation experienced by the observer is never completely identical to the situation of the experienced animal, but while sometimes the differences are minor, other times they are substantial. Moreover, it may not always be obvious what stimulus another individual may be responding to. Proposed mechanisms for social learning need to explain how the correspondence problem may be solved.

## 2.2. Associative learning

Associative learning is commonly recognized as a key element in animals' adaptation to their environment, when innate knowledge is insufficient. For example, associative learning can generate appropriate anticipatory behaviour to stimuli of biological significance, can lead to the discovery of new food sources and can improve foraging efficiency by perfecting innate behavioural programmes [35,36]. Nevertheless, associative learning is sometimes viewed as 'mindless' and unable to produce complex behaviour (e.g. [37–41]). This characterization stands in contrast with recent, dramatic demonstrations in artificial intelligence, in which associative learning systems (coupled with sophisticated perceptual processing and other algorithmic elements) have displayed human-level performance in complex games such as chess, Go and others (e.g. [42–44]). We have previously shown that associative learning, augmented with suitable genetic predisposition, can produce flexible and sophisticated behaviour such as tool use, self-control and appropriate reactions to violations of expectation [28]. In this section, we summarize our model of associative learning, referring to Enquist et al. [28] for a full account and to Sutton and Barto [42] and Wiering [45] for its foundation in artificial intelligence. In the following sections, we will explore its application to social learning.

Social learning phenomena include both learning about stimuli (Pavlovian learning, e.g. predator recognition) and learning about actions (instrumental learning, e.g. learning a foraging technique). Furthermore, social learning includes phenomena in which Pavlovian and instrumental learning interact, such as when learned signals for danger lead to learning about actions that can avoid danger

(§3.5). It is, therefore, paramount to use a model of associative learning that accounts simultaneously for Pavlovian and instrumental learning. In our model, a first equation models learning about the value of stimuli (related to conditioned reinforcement in experimental psychology), while a second equation models learning about the value of actions (stimulus–response (S-R) associations), which is possibly informed by learned stimulus values. Finally, a third equation models how action values are used in decision-making. To introduce the model formally, consider a sequence such as equation (2.1), in which an animal experiences a stimulus S, responds with behaviour B, chosen from the animal's repertoire, and then experiences an outcome $S'$. As a consequence of these experiences, the model updates two quantities. One is $w(S)$, which is the estimated value of stimulus S, i.e. the animal's estimate of how much reward it can expect to collect after experiencing S. The second is $v(S \rightarrow B)$, which is the estimated value of reacting with B to stimulus S, i.e. the animal's estimate of the reward it can collect after responding with B to S. Formally, the memory updates are as follows:

$$\left.\begin{aligned} \Delta v(S \rightarrow B) &= \alpha_v[u(S') + w(S') - v(S \rightarrow B)] \\ \Delta w(S) &= \alpha_w[u(S') + w(S') - w(S)]. \end{aligned}\right\} \tag{2.3}$$

and

The first equation models S-R, instrumental learning. In this equation, the S-R value $v(S \rightarrow B)$ is updated so as to become closer to the total reinforcement value of $S'$. The latter is $u(S') + w(S')$, where $u(S')$ is the innate reinforcement value and $w(S')$ the learned value. This equation is similar to standard error-correction equations such as the Rescorla and Wagner model [46], with the difference that reinforcement values are explicitly partitioned into an innate and a learned part. The second equation updates the stimulus value $w(S)$ in the same way, i.e. bringing it closer to the total value of $S'$. Note that this update depends only on which stimulus $S'$ follows stimulus S and not what response B was performed. In this sense, the second equation models response-independent Pavlovian processes. Parameters $\alpha_v$ and $\alpha_w$ regulate the rate at which the two kinds of memories are updated (learning rates).

The last equation of the model implements decision-making with a 'softmax' rule. Namely, upon experiencing stimulus S, the model chooses behaviour B with probability

$$\Pr(S \rightarrow B) = \frac{\exp(\beta v(S \rightarrow B))}{\sum_{B'} \exp(\beta v(S \rightarrow B'))}. \tag{2.4}$$

Parameter $\beta$ regulates the amount of exploration: if $\beta = 0$ all behaviours are equally likely, whereas if $\beta$ is large only the behaviour with the highest $v$ value will be selected with appreciable probability. In all simulations below, we use values $\alpha_v = 0.1$, $\alpha_w = 0.1$ and $\beta = 1$. All graphs report averages of 1000 simulations, apart from the avoidance learning example that reports average values for 2000 simulations.

A crucial feature of the model is that it enables learned stimulus values ($w$ values) to modify S-R values ($v$ values) and thus to reinforce instrumental responses. In the present scope, this mechanism is important for social stimuli to influence behaviour, as we discuss in §3.4. More generally, stimulus value learning makes the model capable of learning sequences of actions. It can be proved that, given enough experiences, the model learns the correct $v$ and $w$ values for stimuli [28] and thus approximate the optimal course of action.

Social learning situations frequently include complex stimuli composed of both social and non-social elements, such as when observing a conspecific manipulating a fruit or other potential food sources. To handle such situations in the model, we proceed in a similar way as in classic models of associative learning, such as the Rescorla and Wagner model [46]. Namely, we assume that the $v$ value of a compound stimulus is computed as the sum of the $v$ values of its components

$$v(S_1 S_2 \ldots S_n \rightarrow B) = \sum_i^n v(S_i \rightarrow B). \tag{2.5}$$

The same sum rule applies to the $u$ and $w$ values of stimuli, such as in $u(S_1 S_2) = u(S_1) + u(S_2)$. We also assume that an experience with the compound stimulus leads to learning about each of its components, according to

$$\Delta v(S_i \rightarrow B) = \alpha_v[u(S') - v(S_1 S_2 \ldots S_n \rightarrow B)] \tag{2.6}$$

and similarly for $w$ values. These simple assumptions about stimulus compounds are adequate in the cases we discuss below. The current approach can be refined, when needed, to include such factors as stimulus elements with different salience and stimuli that vary along continuous dimensions [28].

## 2.3. Modelling social learning in terms of associative processes

In a model of social learning that seeks to appeal only to general associative processes, social and non-social stimuli are processed and learned about in exactly the same way. Thus, technically, there is nothing special about social learning in such a model [19,25,27]. At the same time, social learning situations are often more complex than better-known associative learning scenarios such as simple Pavlovian and instrumental conditioning. We will see below that an associative account of social learning requires specifying the learning scenario in detail, similarly to how learning of complex behaviour is analysed in the Skinnerian tradition [47–49]. This includes identifying which social and non-social stimuli are experienced, which behaviours can be performed, what are the consequences of behaviour, what are the reinforcement values of stimuli, and so on. Previous experiences may also be important, which in our model are summarized by starting with non-zero initial $v$ values for some S-R pairs, and with initial $w$ values for some stimuli. We will also take into account various kinds of genetic predispositions, as highlighted by the ethological tradition [50–52], and by psychological theories that have sought to frame learning within its biological background [53–55]. Genetic predispositions are likely to play a crucial role in social learning through such mechanisms as attributing high salience to social stimuli and favouring specific responses to social stimuli [25,27,56]. We will see below how such predispositions can be included in our model (see [28, table 2], for a general discussion).

We acknowledge from the outset that associative learning does not directly accomplish some of the most sophisticated social learning faculties. For example, it does not contain a dedicated mechanism to solve the correspondence problem in general. Rather, the correspondence problem is solved mainly on a case-by-case basis, as we will see below, although extensive training may provide some more general abilities (see §4.3). Furthermore, our model does not remember observed behavioural sequences, implying that it cannot immediately imitate unknown sequences. It is unclear, however, whether these limitations imply that associative learning is insufficient to understand non-human social learning, as it is a matter of debate whether animals possess general solutions to the correspondence problem, and whether they can immediately imitate action sequences (cf. [57,58]). It is also uncertain whether natural selection would favour the evolution of learning mechanisms dedicated to social learning, as general learning mechanisms such as associative learning can optimize behaviour in response to social stimuli [28,42]. We do not seek to resolve these issues completely, although we touch upon them in the Discussion. Rather, the examples we present below aim to illustrate how associative learning can provide a mechanism for social learning that is plausible and requires minimal assumptions.

# 3. Results

## 3.1. Diversity of social learning scenarios

The many possibilities for observing social and non-social stimuli imply a wealth of social learning scenarios. Consider, for example, a succession of two stimuli, $S \to S'$, which is the observational part in the elementary sequence $S \to B \to S'$ based on which our model learns (equation (2.1)). Consider further just one social stimulus, $S_{\text{social}}$, one non-social stimulus, $S_x$, and their compound $S_{\text{social}}S_x$. From these stimuli we can form $3 \times 3 = 9$ possible successions, such as $S_{\text{social}}S_x \to S_x$, $S_x \to S_{\text{social}}$, or $S_x \to S_{\text{social}}S_x$. If we consider one more stimulus, either social or non-social, we can form seven combinations, and thus 49 possible successions. Even more possibilities arise if we consider the fact that social learning scenarios often include more than two steps, each of which may or may not include social stimuli [27,59]. The diversity of potential social learning scenarios may partly explain the vast terminology that has been created to describe social learning, and the controversy regarding whether learning in different scenarios is supported by different learning mechanisms.

Below we apply associative learning to five social learning scenarios, summarized in table 1. The first three involve learning of a single response and consider only S-R learning ($v$ values). The other cases consider also stimulus value learning ($w$ values) and explore how the combination of instrumental and Pavlovian learning can support social learning of behavioural sequences and of danger avoidance. In setting the stage for this analysis, we would like to stress that the most impressive cases of social learning concern behavioural sequences rather than single responses. For example, birds that learned to feed from milk bottles in England learned a sequence that included recognizing milk bottles, approaching them, piercing the metal foil cap and finally drinking the cream [60,61]. Learning of tool use can include even longer sequences [28,62]. It is thus important to work with models, like

**Table 1.** A summary of social learning scenarios discussed in §§3.2 to 3.5. The imitation scenario is identical to the previous scenario with $S_{social} = [B]$. The table lists only the most relevant initial conditions, experiences and outcomes. See text and electronic supplementary material for full details. Symbols: $S_{social}$: social stimulus; $S_x$, $S_y$: non-social stimuli; $S_{reward}$: rewarding stimulus; $S_{warning}$: warning stimulus (e.g. warning call); $S_{predator}$: sight, sound, sensation or smell of predator; $[B]$: stimulus corresponding to the learner's perception of response $B$; $B_{ignore}$: ignoring the predator; $v(S \to B)$: S-R value; $w(S)$: learned stimulus value (conditioned reinforcement); $u(S)$: innate stimulus value.

| learning scenario | section | initial conditions | experiences | outcomes |
|---|---|---|---|---|
| response to social stimulus | 3.2 | $v(S_{social} \to B) = 0$ | $S_{social} \to B \to S_{reward}$ | $v(S_{social} \to B) \gg 0$ |
| response imitation | 3.2 | $v([B] \to B) = 0$ | $[B] \to B \to S_{reward}$ | $v([B] \to B) \gg 0$ |
| response to non-social stimulus | 3.3 | $v(S_{social} \to B) \gg 0$ $v(S_x \to B) = 0$ | $S_{social}S_x \to B \to S_{reward}$ $S_x \to B \to S_{reward}$ | $v(S_x \to B) \gg 0$ |
| response sequence | 3.4 | $v(S_{social} \to B_1) \gg 0$ $v(S_x \to B_1) = 0$ $v(S_y \to B_2) = 0$ | $S_{social}S_x \to B_1 \to S_{social}S_y \to B_2 \to S_{reward}$ $S_x \to B_1 \to S_y \to B_2 \to S_{reward}$ | $w(S_y) \gg 0$ $v(S_x \to B_1) \gg 0$ $v(S_y \to B_2) \gg 0$ |
| avoidance learning | 3.5 | $u(S_{warning}) \ll 0$ $u(S_{predator}) = 0$ | $S_{predator} \to S_{warning}$ | $w(S_{predator}) \ll 0$ $v(S_{predator} \to B_{ignore}) \ll 0$ |

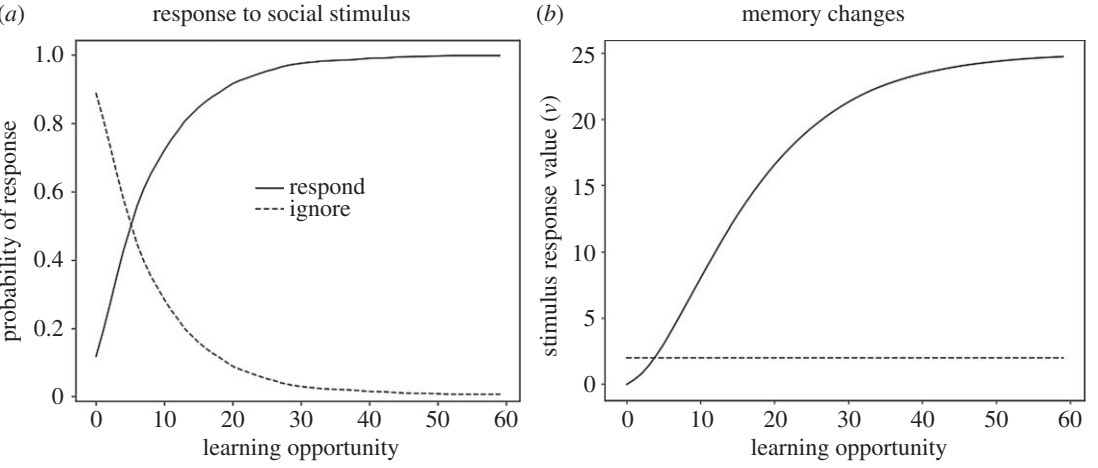

**Figure 1.** Learning to respond to a social stimulus in the situation described in equation (3.1). Panel (*a*) shows how probability of responding changes as a function of number of learning opportunities. Panel (*b*) shows the underlying changes to memory, specifically the values $v(S_{\text{social}} \rightarrow B)$ and $v(S_{\text{social}} \rightarrow 0)$, where ignore stands for behaviour other than B. See electronic supplementary material for simulation code and details.

the one summarized above, that can learn behavioural sequences and that can make testable predictions about how such learning proceeds.

## 3.2. Learning a response to a social stimulus

Learning a response to a social stimulus may not always be considered social learning, but it serves as a useful introduction to modelling social learning with associative processes. Consider a social stimulus $S_{\text{social}}$ that the observer does not respond to initially. This stimulus could be the presence of another individual in a particular context or a behaviour performed by a conspecific. If some action B towards this stimulus is reinforced, the observer will become more likely to respond with B the next time it encounters $S_{\text{social}}$. This simple scenario involves only two kinds of behaviour sequences

$$\left.\begin{array}{l} S_{\text{social}} \rightarrow B \rightarrow S_{\text{reward}} \\ S_{\text{social}} \rightarrow 0 \rightarrow S_{\text{no reward}}, \end{array}\right\} \tag{3.1}$$

and

where 0 summarizes all responses other than B. We can model this acquisition process by standard instrumental learning, using only the first learning rule in equation (2.3) as no learning of $w$ values (Pavlovian learning) is required. Thus, the S-R value of performing B towards $S_{\text{social}}$ is updated according to

$$\Delta v(S_{\text{social}} \rightarrow B) = \alpha_v[u(S_{\text{reward}}) - v(S_{\text{social}} \rightarrow B)]. \tag{3.2}$$

A simulation of this learning process is presented in figure 1. Figure 1*a* shows that the probability of responding is initially small, but not zero. Once responding starts to occur, response probability increases quickly, eventually reaching an equilibrium value close to 1. Figure 1*b* shows the underlying changes to memory. The S-R association $v(S_{\text{social}} \rightarrow B)$ grows and approaches 25, which is the value of the reinforcer, $u(S')$. The S-R association between the social stimulus and not responding, $v(S_{\text{social}} \rightarrow 0)$, remains unchanged as this response is not reinforced.

This account is no different from standard instrumental learning, and the use of any behaviour can be modified in this way. This includes behaviour directed towards another individual such as social, aggressive or sexual behaviour; and other behaviours that may be elicited by the social stimulus. For instance, a young individual may learn to avoid older and stronger individuals if or when the latter behaves aggressively. This can be thought of as either decreased responding towards a negative stimulus (an aggressive conspecific) or increased responding towards a positive stimulus (reaching safety).

Learning to respond to social stimuli can also account for how an animal may learn to imitate single behaviours [27]. The essence of this account is that an observer can associate different responses to different social stimuli. For example, a conspecific that performs two distinct behaviours, $B_1$ and $B_2$, will give rise to two distinct social stimuli, $[B_1]$ and $[B_2]$. An individual that learns to respond with $B_1$

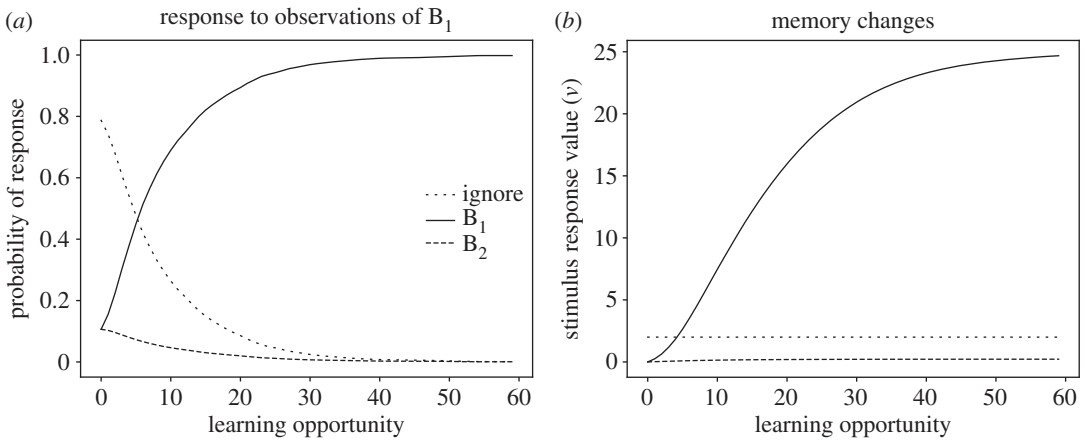

**Figure 2.** Learning to imitate individual behaviours. Social observations contain stimulus elements emerging from two different behaviours, $B_1$ and $B_2$. Repeating the behaviour just observed is rewarded. Panel (a) shows how the probability of responding to observations of $B_1$ changes as a function of number of exposures. Panel (b) shows the underlying changes to memory. The curves for responding to $B_2$ are identical. See electronic supplementary material for simulation code and details.

to $[B_1]$ and with $B_2$ to $[B_2]$ would then be learning to imitate these behaviours, at least from an operational point of view (figure 2). This explanation is consistent with Heyes' [31] remark on imitation in birds, both wild-caught [63,64] and laboratory reared [65,66]. Namely, Heyes concluded that 'the imitated body movements were part of the relevant species' natural foraging repertoire, and the subjects were accustomed to feeding in flocks. Therefore, it is possible that they had learned to imitate these movements during group feeding prior to the experiments. Somewhat surprisingly, the model predicts that animals with smaller behavioural repertoires are more likely to learn to imitate. In fact, all else being equal, the likelihood that an individual will learn to use the same behaviour as another is greater in species with a smaller repertoire, simply because there are fewer behaviours to choose from (see also §3.4 for discussion on combinatorial matters). Animals with large behavioural repertoires, such as primates, may have difficulty in discriminating between similar motor patterns, and will also try out more behaviours during learning, both of which may lead to them eventually adopting a different behaviour than the observed one. A quail or a reptile, in contrast, will have many fewer options in both observations and choice of behaviour. This circumstance may explain the mixed result obtained in imitation studies in primates (e.g. [20,58,67]).

The plausibility of the examples given above depends strongly on whether a suitable reinforcer can be realistically assumed. For example, a pigeon that pecks or scratches the ground when it sees other pigeons pecking or scratching is likely to be reinforced by finding food, and thus can plausibly learn to imitate these behaviours. Similarly, a hummingbird may plausibly learn to avoid flowers where another hummingbird is foraging, because it may find no nectar there or because it may be attacked by the other bird. Explanations of this kind are less immediate when it comes to the imitation of arbitrary motor patterns with no immediate consequences. It should be borne in mind, however, that social stimuli can themselves function as reinforcers, so that behaviour can be reinforced if it leads to being in the proximity of conspecifics or to interacting with them [68–70].

We conclude by noting that the scenario discussed in this section—learning to respond to social stimuli—is probably of limited importance for understanding sophisticated social learning. For example, learning S-R reactions to social stimuli cannot establish a general imitation ability because imitating each behaviour would have to be learned separately. Moreover, in this kind of learning, the learned behaviour is triggered by the presence of the social stimulus, and it will not occur when the learner is on its own. Thus, learning a response to a social stimulus cannot support behaviour when the learner is alone. At the same time, we will see below that responses to social stimuli (learned or innate) can be important to learn responses that are performed when the learner is alone.

## 3.3. Learning to respond to non-social stimuli

Here, we consider how learning triggered by a social stimulus can facilitate the acquisition of behaviour that persists even when the social stimulus is absent. For example, consider a young herring gull that approaches an experienced gull that is feeding on something unknown to the young. If such approach is rewarded, for

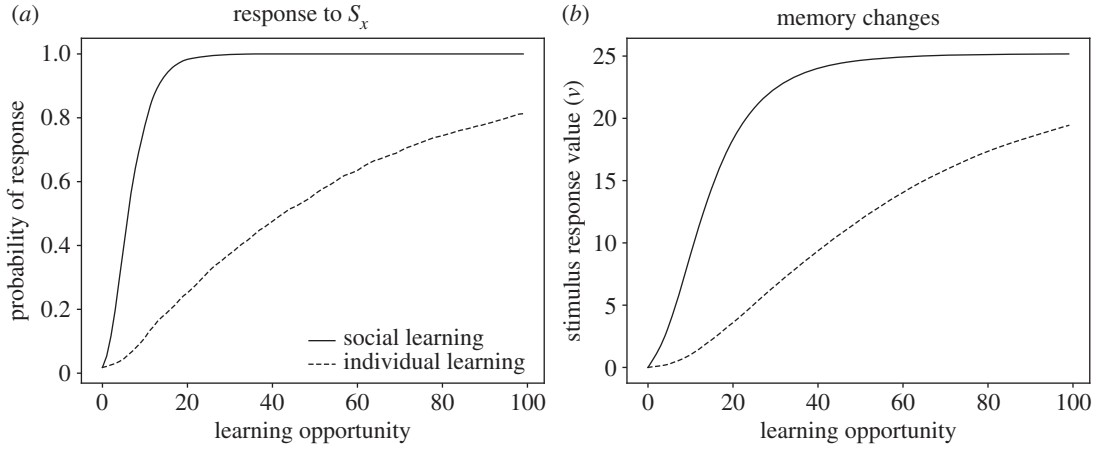

**Figure 3.** Learning to respond to a non-social stimulus through social learning. Panels (*a*) and (*b*) compare social learning (solid line) with individual learning (hatched line). Panel (*a*) shows how probability of responding changes as a function of number of learning opportunities, and panel (*b*) shows the underlying changes to memory. See electronic supplementary material for simulation code and details.

example, by access to the food, it can lead the young to approach the food even in the absence of the experienced individual. A breakdown of this scenario is as follows. The experienced gull provides the social stimulus $S_{\text{social}}$, while the unknown food provides the non-social stimulus $S_x$. When the experienced gull is feeding, the two stimuli are experienced as the compound $S_{\text{social}}S_x$. Let B stand for approach behaviour. We assume that the young initially approaches conspecifics, which we model as $v(S_{\text{social}} \to \text{B}) > 0$. We also assume that the initial value of approaching the unknown food is zero, $v(S_x \to \text{B}) = 0$. When the young gull approaches the experienced one, its experience can be written as

$$S_{\text{social}}S_x \to \text{B} \to S', \tag{3.3}$$

where $S'$ is the consequence of approaching. To calculate the effects of this experience, we use the learning equations for compound stimuli, equations (2.5) and (2.6). Thus, the value of approaching $S_x$ is updated according to

$$\Delta v(S_x \to \text{B}) = \alpha_v[u(S') - v(S_{\text{social}}S_x \to \text{B})] \tag{3.4}$$

$$= \alpha_v[u(S') - (v(S_{\text{social}} \to \text{B}) + v(S_x \to \text{B}))]. \tag{3.5}$$

The tendency to approach the food will increase if $\Delta v(S_x \to \text{B}) > 0$, that is, if the reward experienced after approaching, $u(S')$, is larger than the current value of approaching the compound stimulus $S_{\text{social}}S_x$, that is, $v(S_{\text{social}} \to \text{B}) + v(S_x \to \text{B})$. This may happen, for example, because $S'$ may be perceived as valuable due to proximity to the conspecific, or because after approaching the young bird gains access to the novel food. (Strictly speaking, the latter possibility involves a sequence of actions, which we will consider in the next section.)

The effect of social experiences with $S_{\text{social}}S_x$ is thus to increase $v(S_x \to \text{B})$, which leads to increased approach probability even when the food is encountered in the absence of conspecifics. If social experiences are interspersed with non-social experiences with the food alone, then it will be easier for the young to learn to approach the new food source. We refer to this as *transfer learning* because behaviour learned in a social context transfers to a non-social context. Transfer experiences are important to consolidate responding to X alone. This effect is illustrated in figure 3 when the young encounters $S_x$ alone 80% of the time, and $S_{\text{social}}S_x$ the remaining 20%. This situation is compared to non-social learning in which the young only encounters $S_x$, which results in slower acquisition than in the social learning situation. The effect of social experiences ranges from very significant to negligible depending on model parameters, such as the amount of exploration (determined by $\beta$ in equation (2.4)), and the prevalence of experienced individuals (which determines the probability of observing $S_{\text{social}}S_x$).

These scenarios may apply widely, that is, when social experiences bootstrap learning that subsequently is completed through non-social experiences. For example, instead of learning to approach novel food, an individual can learn to approach a specific location, such as a shelter or foraging grounds. In these cases, the learner may end up using the same behaviour employed by experienced individuals, yet social observations do not directly cause this similarity in behaviour.

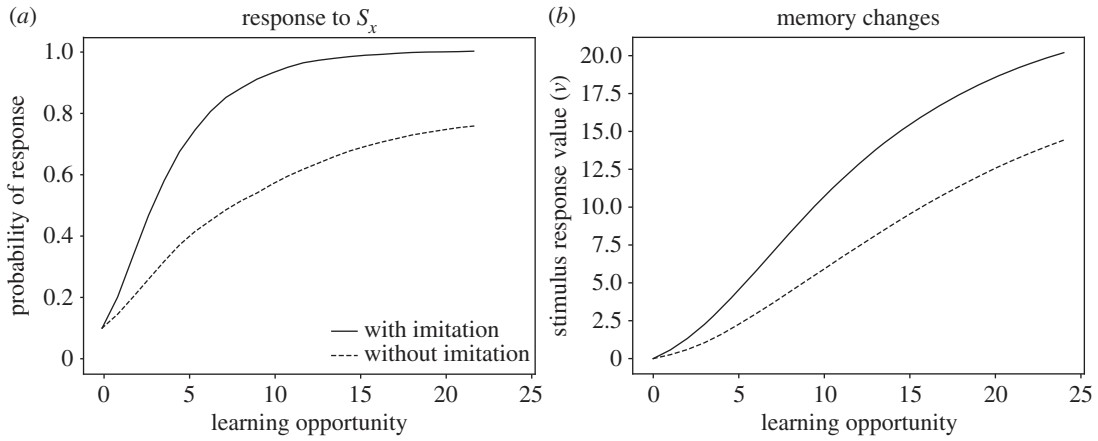

**Figure 4.** Transmission of behaviour though imitation. A learner sees an experienced animal perform a behaviour B in response to stimulus $S_x$. If the learner imitates B, it is more likely to also respond to $S_x$ with B, which facilitates learning when B is rewarded. If the learner does not imitate B, learning proceeds more slowly by simple trial and error. Panel (*a*) shows the probability of responding to $S_x$ with B with and without the ability to imitate B. Panel (*b*) shows the underlying changes to memory. See electronic supplementary material for simulation code and details.

Rather, the role of social observations is to draw the learner in contact with the same reward contingencies as those perceived by the experienced animals. Thus, behavioural similarity is not the product of direct imitation: the learner has still to explore on its own the different behavioural options. Such exploration can be facilitated by genetic predispositions. For example, recognizing that the experienced animal is feeding may bias the learner to use behaviours that are relevant to feeding, among which will be the behaviour that the experienced animal is using.

Learning a response to a non-social stimulus can also be facilitated by pre-existing social responses. Assume that there are 10 possible behaviours, $B_1, \ldots, B_{10}$ and that these can give rise to identifiable stimuli $[B_1], \ldots, [B_{10}]$. Consider now a situation in which an experienced animal uses $B_1$ in response to stimulus $S_x$, such that learners observe the compound $[B_1]S_x$. As before, $S_x$ occurs alone with probability 0.8 and the compound stimulus with probability 0.2. We compare two situations. In one situation, the individual can imitate $B_1$ before the start of the simulation, i.e. they respond with $B_1$ to $[B_1]$. This response may be genetically predisposed or learned as in the preceding section. In the other situation, the individual lacks this ability and instead starts to explore behaviours within its repertoire with equal frequencies. Simulations show, not surprisingly, that productive responding towards $S_x$ is established faster in individuals capable of imitating $B_1$ as shown in figure 4.

## 3.4. Learning a behaviour sequence

A careful examination of social learning phenomena often reveals that the final outcome is a sequence of behaviours guided by a succession of stimuli, rather than a single response to a single stimulus. In one of the examples above, a young gull learned to approach novel food, but in order to benefit from this behaviour the gull must also learn to handle and consume the food. In sequence learning, social learning is often combined with individual learning. For example, in the case of small birds learning to feed from milk bottles mentioned in §3.1 [60], birds were attracted to milk bottles by the presence of other birds, and then learned individually to pierce the foil cap and skim the cream floating atop of the milk [71]. We speculate that many social learning phenomena are variants of this kind of sequence learning, such as stimulus and local enhancement, emulation and opportunity providing (see §4.2). In this section, we explore how social observations can facilitate the acquisition of productive behaviour sequences through associative learning.

There are two main difficulties in learning behavioural sequences. One is that there are many more possibilities to explore than when learning single responses. An animal with a repertoire of $n$ behaviours, for example, can try out $n^l$ behaviour sequences of length $l$. This exponential growth in exploration time as a function of sequence length, easily makes learning too time consuming to be worthwhile. We consider this difficulty towards the end of this section. The second difficulty is that, typically, the initial steps of a sequence are unrewarded. For example, approaching milk bottles is necessary to exploit them as a source of food, but it is not rewarding in itself. Associative learning can solve this problem through

conditioned reinforcement and response chaining [28,36,72,73]. Conditioned reinforcement is a Pavlovian process (in that it is independent of the animal's actions) in which stimuli that anticipate reinforcement acquire themselves reinforcing properties [73,74]. Chaining refers to the fact that single responses can be linked together to form behaviour sequences. In learning to exploit milk bottles, for example, a lucky bird may have happened to perch on the bottle and pecked at the cap, or found an open cap, and thus gained access to the milk. This would have had two consequences: a strengthening of the S-R association between seeing the cap and pecking, and a growth in the conditioned value of seeing the cap. The latter means that, on successive experiences, approaching the cap would have felt rewarding, by virtue of conditioned rather than of primary reinforcement. The conditioned value of seeing the cap would have then reinforced approaching milk bottles.

In our model, conditioned reinforcement takes the form of stimulus value learning ($w$ values in equation (2.3)), and chaining emerges because learned stimulus values act as reinforcers to increase or decrease S-R values ($v$ values in equation (2.3)). In this section, we use the model to exemplify how social observations can facilitate the acquisition of productive sequences. We consider cases in which a sequence of two behaviours is required to obtain a reward

$$S_x \rightarrow B_1 \rightarrow S_y \rightarrow B_2 \rightarrow S_{\text{reward}}. \tag{3.6}$$

For example, this sequence may describe a gull approaching ($B_1$) a food truck ($S_x$) and taking ($B_2$) a paper bag ($S_y$) to feed on the chips ($S_{\text{reward}}$) inside

$$\textit{Food Truck} \rightarrow \textit{Approach} \rightarrow \textit{Paper Bag} \rightarrow \textit{Take} \rightarrow \textit{Chips}.$$

As in the previous section, we assume that a naive bird initially approaches conspecifics ($S_{\text{social}}$), but not $S_x$, the food truck. We also assume that the gull initially does not take $S_y$, the paper bag. Given this set-up, the social learning scenario is as follows. Initially, the naive bird is attracted to the truck by the presence of conspecifics. That is, stimulus $S_{\text{social}}S_x$ elicits approach ($B_1$) because the S-R value $v(S_{\text{social}} \rightarrow B_1)$ is high. Thus, the naive gull experiences the sequence $S_{\text{social}}S_x \rightarrow B_1 \rightarrow S_{\text{social}}S_y$. This experience strengthens the tendency to approach the truck. Formally, $v(S_x \rightarrow B_1)$ increases because $S_{\text{social}}S_x \rightarrow B_1$ is followed by the rewarding stimulus $S_{\text{social}}S_y$ (see §3.3). In turn, approaching the truck often may lead the learner to autonomously learn the response $S_y \rightarrow S_{\text{reward}}$. Thus, S-R learning can strengthen both approaching the truck and taking the paper bag. It is crucial to note, however, that S-R learning is not enough for autonomous performance of the whole sequence. In fact, in the absence of conspecifics, the first step, $S_x \rightarrow B_1 \rightarrow S_y$ is not rewarding. Performing this step would, with time, lead to a decrease of $v(S_x \rightarrow B_1)$, and the gull would stop approaching the truck. With conditioned reinforcement, on the other hand, $S_y$ itself can become a rewarding stimulus, and thus continue to reward approach to the truck even in the absence of social rewards. There are, in fact, two kinds of experiences that endow $S_y$ with conditioned reinforcement value. First, the gull can itself perform $S_y \rightarrow B_2 \rightarrow S_{\text{reward}}$. This causes an increase in $w(S_y)$ according to equation (2.3),

$$\Delta w(S_y) = \alpha_w[u(S_{\text{reward}}) - w(S_y)].$$

Second, when in proximity of the truck, the learner can witness experienced gulls taking paper bags and feeding on the chips, corresponding to the sequence of stimuli $S_{\text{social}}S_y \rightarrow S_{\text{social}}[B_2] \rightarrow S_{\text{social}}[S_{\text{reward}}]$. Embedded in this sequence is the succession $S_y \rightarrow S_{\text{social}}[B_2]$ which endows $S_y$ with conditioned value, again according to equation (2.3),

$$\Delta w(S_y) = \alpha_w[u(S_{\text{social}}) - w(S_y)],$$

where, for simplicity, we have assumed that perceiving $B_2$ is not rewarding, i.e. $u([B_2]) = 0$. Once $S_y$ has acquired conditioned value, it can reinforce and maintain approach to $S_x$ even in the absence of social stimuli. According to equation (2.3), the full learning equation for $v(S_x \rightarrow B_1)$, when the learner is alone, is

$$\Delta v(S_x \rightarrow B_1) = \alpha_v[u(S_y) + w(S_y) - v(S_x \rightarrow B_1)],$$

meaning that $v(S_x \rightarrow B_1)$ is driven towards the positive value $w(S_y)$, even if $S_y$ is not a primary reinforcer ($u(S_y) = 0$). We show in figure 5 that this scenario can lead to an impressive speed-up of learning, compared to the case in which there are no social stimuli. Learning of the sequence could have proceeded even more quickly, if the learner had previously learned to imitate $B_2$, for example, via the route explored in §3.2. In the simulation, the learner encounters $S_x$ alone (food truck without experienced gulls) 80% of the time and $S_{\text{social}}S_x$ (food truck with other gulls) the remaining 20% of the time. The social experiences bootstrap learning, while the individual experiences consolidate responding

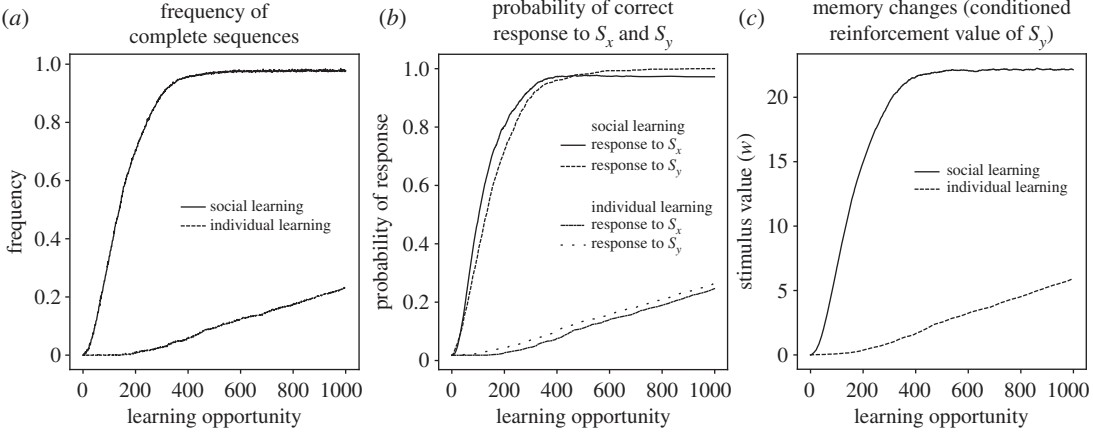

**Figure 5.** Social learning versus individual learning of a behaviour sequence. Panel (*a*) shows the probability of performing the complete correct sequence starting from $S_x$ (see text and table 1) as a function of number of learning opportunities, comparing social and individual learning. Panel (*b*) shows that, in individual learning, the response to the second stimulus, $S_y$, is acquired before the response to the first stimulus, $S_x$ (backward chaining). In social learning, however, the order is reversed because social stimuli facilitate the response to $S_x$. Panel (*c*) shows that social learning also lead to faster growth in $w(S_y)$, the stimulus value of $S_y$, which can further reinforce the correct response to $S_x$. See electronic supplementary material for simulation code and details.

to $S_x$ and $S_y$ in the absence of social stimuli. As noted in §3.3, individual learning experiences are important to achieve a full transfer of the newly acquired skill from a social to a non-social context.

In summary, here social stimuli facilitate learning because they promote the build-up of conditioned reinforcement to intermediate stimuli in the sequence, and because they initially attract the animal towards a situation (the food truck) that would otherwise be ignored. The first factor amounts to modifying the reward structure of the sequence, so that even steps that do not lead to primary reinforcement can be reinforced. The second factor is an example of what we have labelled an 'entry pattern' (see [28], for details), which refers to where in a sequence an animal is likely to find itself. Consider a generic sequence

$$S_1 \rightarrow B_1 \rightarrow \cdots \rightarrow S_l \rightarrow B_l \rightarrow S_{\text{reward}}.$$

If the animal always starts from the first step, and if no rewards are perceived until the sequence is complete, it takes an average of $n^l$ trials to stumble upon the reward for the first time, where $n$ is the number of behaviours to choose from and $l$ is the number of sequence steps. Because $n^l$ grows exponentially with $l$, this entry pattern makes it nearly impossible to learn sequences longer than a few steps. In contrast, it is easiest to learn the sequence backwards by starting at $S_l$, in which case, each step takes about the same time to learn and the total learning time is linear in the length of the sequence. This strategy is often used by animal trainers [75]. In nature, intermediate entry patterns are common, in which sequences are entered from a mix of steps. In a famous case involving social learning, black rats learn to open pine cones by having access, through their mother, to pine cones at different stages of opening, such as cones with partly or fully exposed kernels. Rats that only encounter whole cones, by contrast, never learn the feeding sequence [76–78]. More generally, parents or other individuals can create favourable entry patterns by exposing learners to intermediate steps in a sequence, which would be hard for the learner to reach on its own. Several terms that are commonly used to describe social learning, such as local and stimulus enhancement, emulation and opportunity providing, refer at least in part to ways in which social observations create different entry patterns. For a full understanding of how various social situations can favour the learning of behavioural sequences, simulations such as the one in figure 5 can be conducted.

## 3.5. Avoidance learning

In many species, predator recognition and avoidance is learned based on social signals such as warning calls from conspecifics (for reviews, see [26,79]). Many authors have attributed this kind of social learning to the establishment of an association between the predator stimulus and the warning stimulus [27,80,81], but we are not aware of computational models of social avoidance learning. We first note that learning to avoid

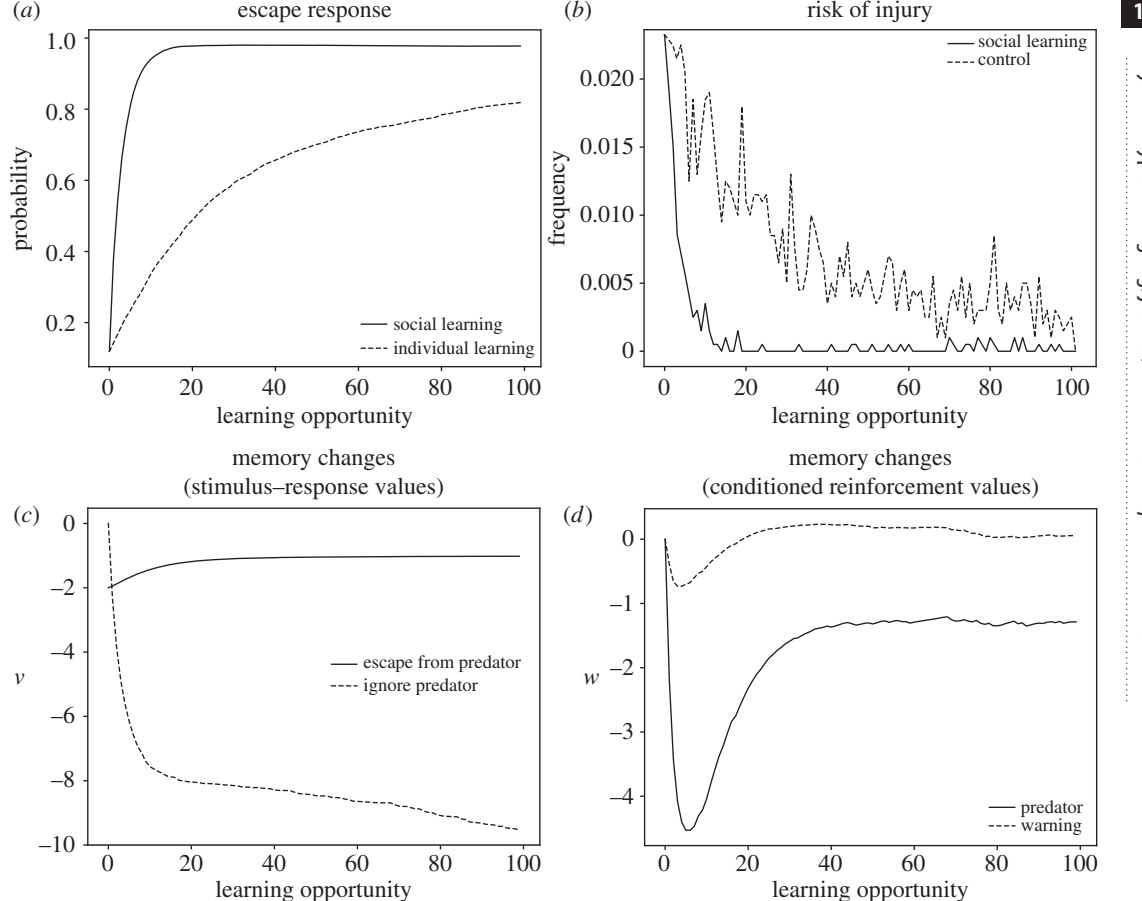

**Figure 6.** Social learning of predator avoidance through inborn recognition of a warning call. (*a*) The probability of escaping the predator in social and individual learning. (*b*) The risk of injury as a function of number of predator encounters comparing social and individual learning. (*c*) The underlying changes to memory for the social learning case. (*d*) Changes in conditioned reinforcement values. See electronic supplementary material for simulation code and further details.

danger does not allow for much error. A modification of the life/dinner principle [82] can illustrate the special problem of learning about lethal stimuli; a young rabbit must learn at once to run for its life while a young fox, however, can refine its dinner catching techniques over repeated attempts. For this reason, we expect genetic guidance to be more important in predator recognition than, for example, in food recognition. Here, we assume that the genes guide learning by endowing the warning stimulus with a large negative value, $u(S_{\text{warning}}) \ll 0$, while the predator stimulus is initially neutral, $u(S_{\text{predator}}) = 0$. The details of our simulation are as follows. In encounters with the predator, the learner is alone half of the time and with an experienced animal the other half. The experienced animal, in turn, gives a warning call half of the time it spots a predator. At each time step, one of three things may happen: the predator leaves on its own (probability of 20%); the learner escapes through a suitable behavioural response, such as burrowing; the learner ignores the predator. In real life, ignoring the predator may lead to injury or death, but we do not model these possibilities explicitly because we are mainly interested in comparing the speed of learning in the presence and absence of warning calls.

The results are illustrated in figure 6 and shows that inborn recognition of conspecific warning calls can facilitate fast transmission of predator recognition between individuals. The key learning event is when the learner ignores the predator and then hears the warning call, corresponding to the sequence $S_{\text{predator}} \rightarrow B_{\text{ignore}} \rightarrow S_{\text{warning}}$. In this event, the negative value of $S_{\text{warning}}$ causes the association between $S_{\text{predator}}$ and $B_{\text{ignore}}$ to decrease, thus decreasing the probability that the predator will be ignored in the next encounter. Moreover, in this event, the stimulus value $w(S_{\text{predator}})$ becomes negative because it precedes a stimulus, $S_{\text{warning}}$, with negative value. This means that $S_{\text{predator}}$ becomes capable of reinforcing escape behaviour, because ignoring the predator results in the experience $S_{\text{predator}} \rightarrow B_{\text{ignore}} \rightarrow S_{\text{predator}}$, in which the negative value of the (second) predator stimulus reduces the S-R value of ignoring the predator. Individual learning of avoidance is much slower because nothing can be learned unless the predator

attacks. In the simulation, the attack simply punishes ignoring the predator, but in real life, the individual may die before learning to avoid the predator.

Note that stimuli other than $S_{predator}$ may become associated with the warning. For example, superb fairy-wrens (*Malurus cyaneus*) can learn the warning calls of other species by associating them with the conspecific warning call [83]. Such learning can occur if learners experience sequences such as $S_x \rightarrow B_{ignore} \rightarrow S_{warning}$, where $S_x$ is the heterospecific warning call. The two calls, heterospecific and conspecific, are likely to occur in this sequence at least some of the time, because they are both triggered by the presence of the predator. If these experiences recur, $S_x$ eventually acquires conditioned value equal to the value of $S_{warning}$. That is, the outcome of learning is $w(S_x) = u(S_{warning})$. This conditioned value can then punish ignoring the predator when the heterospecific warning call is heard, that is, when the sequence $S_{predator} \rightarrow B_{ignore} \rightarrow S_x$ is experienced. Thus, the learner now 'understands' heterospecific as well as conspecific warning calls.

Our simulation is certainly simplified, yet offers an example of how associative learning may lead to predator recognition based on a warning signal. In reality, avoidance of dangerous stimuli is probably supported by further genetic predispositions. For example, it is important that the learner becomes afraid of the predator and not of other surrounding objects, such as conspecifics or food items, that may be present when the predator appears. This may, in principle, derive from discrimination learning, because innocuous objects will not give rise to warning calls, but discrimination learning is often time consuming. An alternative is that genetic predispositions bias learning towards stimulus features that are probably diagnostic of predators, such as movement or a certain size. In addition, there may be genetic predispositions such that the warning call itself has direct effects on behaviour, such as increasing vigilance or priming a flight response.

# 4. Discussion

We have studied a few learning scenarios to demonstrate how social learning may arise based on a combination of associative learning and genetic predispositions. Our analysis of social learning, like other associative analyses (e.g. [18,27]), contrasts with descriptions of social learning that make no reference to associative processes. Here, we explore whether these descriptions may be, in fact, amenable to associative analysis. For definiteness, we consider the descriptive terms in table 2, which is based on a recent, comprehensive monograph on social learning [11, table 4.1]. We divide these terms into three groups: those referring to phenomena that can be explained by assuming prior learning or genetically predisposed behaviour; those referring to phenomena that can be explained by learning of S-R ($v$ values) and stimulus values ($w$ values); and those referring to phenomena that are more difficult to explain with our associative learning model. The terms in table 2 are listed in the order in which they are covered below.

## 4.1. Terms accounted for by initial values or genetic predispositions

While potentially general, the term *inadvertent coaching* appears to have been used only for one kind of observation, in which male brown-headed cowbirds (*Molothrus ater*) learn to modify their song based on feedback from females [84,85]. In our model, such learning is possible if female behaviour acts to reinforce the performance of some songs or song elements, but not others, thereby changing the S-R values underlying song choice. In this case, the stimulus is the female, or some behaviour by the female, and the response is the male's song. Genetic specializations would determine the reward value of female reactions to song, and, possibly, female responses to male song.

Terms *social facilitation* and *response facilitation* are often used without directly referring to learning. In these cases, a particular behaviour is described as arising from the presence of other individuals or as a response to the behaviour of another individual. For example, an animal may eat more in the presence of others (see [86], for more examples). In our model, these outcomes can arise either as a consequence of genetic predispositions or because of prior experience. Thus, frequently feeding with conspecifics can result in the tendency to feed at a higher rate when in the presence of conspecifics, or in imitating specific behaviours such as pecking in birds, because such behaviour can be rewarded with food. Social facilitation is simpler to implement than response facilitation, because the naive individual only needs to respond to the presence of another individual rather than to the other individuals' behaviours. The hypothesis that social and response facilitation arise from associative learning provides a useful empirical perspective to study these phenomena. For example, it predicts that social and response facilitation should covary with the amount of experience with the situation that elicits the behaviour.

**Table 2.** Some terms used to describe social learning, adapted from Hoppitt & Laland [11]. Transfer learning refers to additional learning in a non-social context. In observational conditioning and observational S-R and R-S learning, S stands for stimulus and R for response.

| descriptive term | definition | requires transfer learning | accounted for by associative learning |
| --- | --- | --- | --- |
| inadvertent coaching | feedback from experienced animal modifies learner's behaviour | no | yes |
| social facilitation | presence of experienced animal triggers learner behaviour | no | yes |
| response facilitation | behaviour of experienced animal triggers similar behaviour in learner | no | yes |
| contextual imitation | learner copies a familiar action displayed by experienced animal | no | yes |
| stimulus enhancement | behaviour of experienced animal causes learner to learn about a stimulus | yes | yes |
| local enhancement | behaviour of experienced animal causes learner to learn about a location | yes | yes |
| opportunity providing | behaviour of experienced animal creates favourable conditions for learning | yes | yes |
| emulation | learner uses outcomes of experienced animal's actions to learn, but does not copy actions of experienced animal | yes | yes |
| observational conditioning | observations of experienced animal's behaviour change S-S associations in the learner | yes | yes |
| social enhancement of food preferences | food preferences are learned from experienced animal | yes | yes |
| observational S-R and R-S learning | observations of experienced animal's behaviour change S-R or R-S associations in the learner | yes | no |
| production imitation | learner copies one or more unfamiliar actions displayed by experienced animal | yes | no |

Similarly to social and response facilitation, the term *contextual imitation* refers to observations of behaviour rather than to how the behaviour is learned. As seen in §3.2, contextual imitation may be learned by associating the perception of a behaviour with the performance of the same behaviour, resulting in such behaviours as pecking when others are pecking, or raising an arm when the experimenter raises an arm. That contextual imitation can arise from associative learning is compatible with it being taxonomically widespread [63,65,87–89]. Note, however, that our model can only repeat actions one by one. It cannot first observe a sequence of more actions and then repeat it, because it has no memory of past stimuli. We consider imitation of behavioural sequences in §4.3 below.

## 4.2. Terms accounted for by stimulus–response and stimulus value learning

Terms such as stimulus enhancement, local enhancement, opportunity providing and emulation represent more interesting cases of social learning because they refer to situations in which animals, based on social experiences, can learn new productive behaviour that they can later enact on their own. Based on our analysis, it appears that these terms refer to phenomena within the scope of associative learning.

*Stimulus and local enhancement* describe situations in which a social stimulus affects the behaviour by directing exploration towards a non-social stimulus or a location, respectively. In our model, this can

arise through transfer of value between the social stimulus and the non-social stimulus, as seen in §3.3. We have assumed a primary reinforcer value for approach and exploratory behaviour towards social stimuli. It is of course also possible that social stimuli, and responses to social stimuli, have acquired high values before learning takes place through prior experiences; for example, through early interactions with parents and other relatives. In *emulation*, the experienced animal obtains an outcome that the learner recognizes as valuable, such as food. This can facilitate learning in two ways. First, it exposes the learner to a situation in which the reward is more apparent and/or it can be obtained more easily. Second, perceiving the reward can motivate behaviour that is generally useful to obtain that reward, such as exploration and manipulation in the case of food (these effects can be included in our model as context-dependent action selections, see Enquist *et al.* [28]). Once these facilitations are in place, learning can proceed through associative learning (§3.4). Similar arguments hold for *opportunity providing*, in which the experienced animal creates a situation conducive to learning. Opportunity providing can substantially speed up learning and at the same time remain within the scope of associative learning. For example, many cat species bring home live prey to their young to practice predation. Meerkats are even more sophisticated, as they bring to their young scorpions that are progressively less disabled depending on the age of the young [90]. While this behaviour may be labelled 'teaching' on the part of the parent, it is still compatible with the young using associative learning.

Our model can also account for *observational conditioning* through the transfer of value from innately recognized stimuli to other stimuli. For example, warning calls [91,92], warning substances [93] and predator odours [94] often elicit innate anti-predator behaviour. In social learning of predator recognition, these innately recognized stimuli co-occur with the stimuli to be learned about, namely, the sight, sound or smell of predators, and can thus reinforce learning of predator avoidance (§3.5). This scenario appears consistent with empirical examples of predator recognition, such as in blackbirds (*Turdus merula*, [1]), rhesus macaques (*Macaca mulatta*, [81,95]) and prairie dogs (*Cynomys ludovicianus*, [96]). *Social enhancement of food preferences* can also develop through exposure to compound stimuli, assuming that experiencing a novel smell on a familiar individual reinforces exploration of the smell when experienced on its own.

While some of our simulations tracked only S-R values for simplicity, stimulus value learning is equally important in many social learning scenarios. For example, when a young chimpanzee observes her mother using stones to crack nuts, her experiences include closed nuts and stones predicting the availability of open nuts. We thus expect closed nuts and stones to acquire conditioned value [97], and thereby the ability to reinforce exploration and manipulation of nuts and stones that is conducive to mastering nut cracking [28]. The predator recognition scenario discussed above provides another example of the potential role of stimulus value learning: as predator recognition progresses, stimuli associated with predators are predicted to acquire negative value, and thus can reinforce avoidance of locations or stimuli that are correlated with encountering predators.

## 4.3. Terms not accounted for by our associative learning model

There are two main social learning phenomena that are commonly discussed, but which our model does not exhibit. The first is *observational learning* of S-R and response–stimulus (R-S) associations. The second is *production imitation*, also called *true imitation*, which refers to learning new behavioural skills purely by observation [8,98,99]. In our model, observational learning is not possible because S-R associations ($v$ values) can change only when the learner itself performs the response. Merely observing an experienced animal performing a response does not change S-R associations in the learner. Our model, however, can produce a weaker form of observational learning, in which observing the sequence $S_x \rightarrow$ B $\rightarrow S_{\text{reward}}$ performed by another individual can result in faster learning, albeit not immediate. For example, suppose that the learner has already learned to imitate B and assume further that observation of the sequence $S_x \rightarrow$ B is perceived by the learner as the compound stimulus $S_x$[B], for example, because $S_x$ and [B] partly overlap in time. Because of the established imitation, the compound $S_x$[B] will evoke B in the learner. If B is followed by $S_{\text{reward}}$, then the learner will have experienced the sequence $S_x$[B] $\rightarrow$ B $\rightarrow S_{\text{reward}}$, resulting in the growth of the S-R association $v(S_x \rightarrow$ B) as seen in §3.3. Additionally, observations of the form $S_x$[B] $\rightarrow S_{\text{reward}}$ results in $S_x$ and [B] acquiring stimulus value, and thus reinforce exploration, approach and other behaviour that may be conducive to learning B in response to $S_x$.

Turning to observational learning of R-S associations, we note first that these associations refer to the knowledge that a given behaviour (R) results in a given outcome (S). Because the outcomes of behaviour depend typically on the situation, these associations are perhaps better described as S-R-S′ information, that is, knowledge of the outcome (S′) of a response (R) to a stimulus (S). Our model learns about the

value of responses, but not about their specific outcomes. Distinguishing between knowing the value of responding to S with R and knowing the full S-R-S′ information is complex and related to the open question of what associations are formed during instrumental learning [15,35]. We leave this issue to future work, and here we simply remark again that our model does not learn responses by mere observation. Should compelling evidence for observational learning be gathered, the model would need to be revised.

*Production imitation* is more complex than observational learning because it may involve sequences of actions rather than single actions. Thus, in addition to observational learning of actions, production imitation requires a faithful memory of sequences of observations. The extent to which animals are capable of production imitation is debated and resolving this issue is not our present aim. Some studies with chimpanzees [20,67,100], parrots [101] and dogs [102] have demonstrated some degree of motor imitation after extensive training, but the fidelity of imitation is not high. That imitation of behavioural sequences should be difficult for animals is suggested by studies of working memory, indicating that memory for stimulus sequences is much poorer in most non-human species than in humans [103]. In our model, social observations can facilitate behavioural sequence learning (§3.4), but immediate imitation is prevented by the lack of observational learning and of a memory for sequences.

The debate on both production and contextual imitation is intimately tied with the correspondence problem (§2.1). Our model does not contain a dedicated mechanism to solve the correspondence problem (§2.3), yet we saw in §3.2 that the ability to imitate can be learned for specific behaviours. It is possible to develop this account to obtain a more general ability to imitate, by assuming that extensive training can establish very many specific imitative responses, which may form the basis for imitating novel behaviours through stimulus and motor generalization. Stimulus generalization may enable the parsing of observations of novel movements in terms of known motor elements, and motor generalization may enable the construction of novel movements by assembling the known motor elements that have been observed. For example, let $B_1$ stand for 'raise arm' and $B_2$ for 'open hand'. Learning to imitate these behaviours would lead to stimulus $[B_1]$ evoking $B_1$ and to stimulus $[B_2]$ evoking $B_2$ (§3.2). Then the observation of another individual simultaneously raising her arm and opening her hand would give rise to the compound stimulus $[B_1][B_2]$, which through generalization could recruit both responses $B_1$ and $B_2$. While a full computational implementation of this idea is not trivial, this perspective is compatible with the finding that learning of general imitation, to the extent that it succeeds, requires extensive training [20,100–102]. A testable prediction of this account is that imitation abilities, even if extensive, should be limited to novel movements that can be effectively decomposed in terms of trained movements.

## 4.4. Conclusion

Our results illustrate that a diversity of social learning phenomena can arise from associative mechanisms, as formalized in our genetically guided associative learning model [28]. Associative learning may thus provide a unified account of social learning phenomena that are often treated as separate. Our results also indicate that social learning may rely on the same mechanisms as individual learning, with adaptive specializations for social learning implemented as genetic predispositions in perceptual, motivational and reward systems, rather than as distinct learning mechanisms. This perspective is appealing from an evolutionary point of view, as it makes it possible for social learning to evolve by fine-tuning associative learning to a species' social and environmental circumstances, without requiring the evolution of new learning mechanisms. For example, it has been suggested that dogs have evolved to find social stimuli more salient and rewarding than their wolf ancestors [104,105]. Thus, it may not be necessary for social species to have specific social learning mechanisms, as members of social species have naturally more social learning opportunities than members of solitary species. An associative perspective on social learning also suggests that animals do not inherit social learning strategies such as 'copy successful individuals' [106], but that such strategies may be learned based on environmental circumstances [107]. For example, an animal may learn to copy the best foragers because doing so is reinforced more often than copying poor foragers.

Our conclusions are similar to those reached by Cecilia Heyes, whose work has been an important source of inspiration for us [25,27,107]. For example, our account of imitation as learned S-R associations is an implementation of Heyes' approach to the correspondence problem. The main difference between Heyes' work and ours is that our model contains fewer learning processes. For example, Heyes' theory of social learning includes observational learning [25,27], while in our model responses can be learned only by performing them. A similar remark applies to Heyes and Ray's [108] associative sequence learning model (see also [31,109]). In this model, the correspondence problem is solved by learning associations

between behaviours and perceptions of behaviour, as we explored in §3.2. A behavioural sequence can then be learned by remembering the sequence of perceived behaviours and performing for each one the associated behaviour. We have not included this mechanism in our model because sequential memory appears poor in most animals [103], and because we wished to explore a simpler alternative (§3.4). Even without observational learning and a sequential memory, our model is powerful enough to learn relatively long behavioural sequences, such as those seen in primate tool use [28].

More generally, our model holds that behavioural sequence learning depends on stimulus value learning and S-R learning, in concert with genetic predispositions. Overall, this picture agrees with both traditional analysis of behaviour chains in behaviourist psychology [48,72] and with data from more naturalistic settings. For example, Tomasello [97] notes that chimpanzees learn readily about the value of objects manipulated by others, but not about the specific behaviours that others perform. Likewise, in her pioneering work on animal culture, Goodall ([110, p. 561]) summarized chimpanzee learning of tool use as 'a mixture of social facilitation, observation, imitation, and practice—with a good deal of trial and error learning thrown in'. Given that imitation of specific acts can arise from associative learning (§3.2), Goodall's account is close to our conclusions. We emphasize that it is not possible to discriminate between our model and alternative accounts without knowledge of an animal's prior experiences ([15, p. 306] and [31,60]). For example, an imitative behaviour might be genetically predisposed, it might be learned based on observation, or it might be learned by standard instrumental learning as in our model (§3.2). As the final outcome is identical (imitation), only information about prior experience can distinguish between these possibilities.

To end, we concur with Heyes [27] that 'the suggestion that social learning is mediated by associative processes does not imply that all learning is associative'. In animals, exceptions exist in systems with strong genetic support, such as in song learning in hummingbirds, passerines and parrots [111]. In human social learning, mechanisms beyond associative learning have been suggested numerous times (e.g. [112,113]). In both cases, quantitative associative models can provide at least a null hypothesis to test the existence of other social learning mechanisms. The study of social learning naturally touches upon crucial issues in animal cognition, behavioural ecology, development and learning theory, providing numerous exciting opportunities for rigorous study using formal learning models, whether associative or not.

Data accessibility. Data and graphs from our simulations can be generated using software and code as specified in the electronic supplementary material.

Authors' contributions. J.L., S.G. and M.E. conceived the research, performed computer simulations, analysed the data and wrote the manuscript. All authors gave final approval for publication.

Competing interests. We declare we have no competing interests.

Funding. J.L., S.G. and M.E. have been supported by grant no. 2015.0005 from the Knut and Alice Wallenberg Foundation. S.G. was additionally supported by a CUNY Graduate Center fellowship from the Committee for Interdisciplinary Studies.

Acknowledgements. The authors thank Knut and Alice Wallenberg Foundation for support. We also wish to thank three anonymous reviewers.

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
