## [Reviewer comments · Royal Society Open Science]

Review History

RSOS-181777.R0 (Original submission)

Review form: Reviewer 1 (Marco Smolla)

Is the manuscript scientifically sound in its present form?

Yes

Are the interpretations and conclusions justified by the results?

Yes

Is the language acceptable?

Yes

Is it clear how to access all supporting data?

Yes

Do you have any ethical concerns with this paper?

No

Have you any concerns about statistical analyses in this paper?

No

Recommendation?

Accept with minor revision (please list in comments)

Comments to the Author(s)

My thanks to the Editor and the Authors for the opportunity to review the present paper.

Social learning theory has grown dramatically in recent years. One focus has been its underlying cognitive mechanisms. The submitted paper explores how social learning phenomena can be explained by a combination of genetic predispositions and associative learning. The Authors introduce a simple model to formalise associative learning of (social, non-social, or a mix of both) stimuli that covers a surprising range of social learning terms. I find the paper compelling. Just as minimal neural networks give an idea of how complex a mechanism has to be to solve a given problem, the present model gives insight into the minimal requirements for solving a given social learning situation. Some specific comments follow, most of which are minor.

1. ll. 53–56 While I generally agree with the statement, it appears to be a bit too general. Copying the majority, for example, can be seen as social learning but does not require the majority to be experienced (e.g. mate-choice copying in *Drosophila*). Social learning can also be useful for updating information (e.g. waggle dance communication in honey bees).

2. l. 100 "... if a sequence of behavior is considered rather than a single a response.": Is this sentence supposed to end with "behavior"?

3. l. 257 Doesn't the stimulus-response association approach 25 in figure 1?

4. ll. 277–278 "Somewhat surprisingly, the model predicts that animals with smaller behavioral repertoires are more likely to learn to imitate.": Where is this shown?

5. Following l. 315 eq. (9) refers to the non-social stimulus as S_X , whereas in the text above and below it seems to be referred to as X only. Is there a difference between the two? Personally, S_X might be the better option for a non-social stimulus as it follows the nomenclature of S_{social} . Maybe it is worth changing it in the rest of the manuscript.

6. l. 371 "... sequences of l behaviours.": Should this be "of length l "?

7. l. 441 "sequence backwards by starting at S_n " If the sequence started from the end should it then start with S_1 ?

8. l. 454 "simulations such as the one in 5 can be conducted." Should that be a reference to section 3.5?

Review form: Reviewer 2**Is the manuscript scientifically sound in its present form?**

Yes

Are the interpretations and conclusions justified by the results?

Yes

Is the language acceptable?

Yes

Is it clear how to access all supporting data?

Yes

Do you have any ethical concerns with this paper?

No

Have you any concerns about statistical analyses in this paper?

No

Recommendation?

Accept with minor revision (please list in comments)

Comments to the Author(s)

Review of RSOS-181777

This paper describes a model for associative learning taking into account various mechanisms and sequences of behaviours and stimuli from a social perspective. I think the writing was fantastic and have very few comments. My one confusion by the end of it was the applicability or usefulness of the model for those of us that study learning. The models were created and run, giving a nice overview of expectations I guess under certain learning scenarios. However I think that the purpose of the models were a little bit muddled by so much discussion. I would just suggest to make the aim of the model, and its potential applicability to future studies much more clear throughout the paper. In other words, why bother doing this, or why should others bother thinking about it? Can it help us just to understand? Or does it give us a really nice theoretical framework in which to interpret our own experimental results? (I would suggest both). I think just reinforcing the value of this computation model and its potential to help reserchers analyse, interpret and discuss their own work in context would help the paper reach its full potential. Other than this, I think it's wonderful work and I really appreciate the effort that has gone into this!

Minor comments

Line 25 - I'm not sure what you mean by acquiring "productive behaviour" - is there another way to describe what you are trying to say? Can you clarify?

Line 26 - One of the major benefits of social learning is reducing individual risk (rather than, say, energy expenditure). I'd suggest mentioning this here.

Line 29 - You should probably provide a short definition of what you mean by traditions and/or culture here, considering you're just trying to introduce the whole topic your terms need to be very clearly defined.

Lines 64-65 - You say "as a benchmark" twice in the sentence.

Line 59 (and elsewhere) - Can you just define the aim of your model a little bit more? Your title implies it will be computational, will it therefore be predictive or simply explanatory? Do you hope it will be used by others as a framework in which to analyse their own data or experiments?

What are you hoping is the outcome from introducing this model, how will this model serve the field specifically? Just a bit more here would be great.

Line 100 – Remove “a” before the word response.

Line 167 – This is great, a really important feature allowing for individual actions to reinforce (or not) particular learned behaviours.

Line 637 – I would say that there is an exception to this rule in birdsong: most studies have used production imitation as the explanatory process of learning to sing (either through active tutoring or eavesdropping, or even copying non-biotic sounds somehow, such as in Superb lyrebirds or mockingbirds). It is thus widespread in passerines in this particular context (although they need a complex set of brain structures and a highly developed auditory and vocalization pathway for this) – the sequences are based on auditory memories (but take a lot of self-reinforcement). I would probably mention this because song is not mentioned much in this paper yet this is a major theme for some of us that study learning and cognition. It’s an interesting one because while there is a reward, this reward only comes after a very long time, once the energy has been put in (for months or years sometimes) of imitation and production. The motivations are still very unclear, but thought to have a genetic component.

Review form: Reviewer 3

Is the manuscript scientifically sound in its present form?

Yes

Are the interpretations and conclusions justified by the results?

Yes

Is the language acceptable?

Yes

Is it clear how to access all supporting data?

Yes

Do you have any ethical concerns with this paper?

No

Have you any concerns about statistical analyses in this paper?

No

Recommendation?

Accept as is

Comments to the Author(s)

This is another in a great series of papers showing how certain behaviours that are thought to require specialized and/or complex cognitive mechanisms can be produced via associative learning. I really enjoyed reading the paper, and thought you did an excellent job of showing how your model could account for many different kinds of social learning in a clear and accessible way.

One (very) minor but valuable point that needs to be made over and over is that the "deep

learning" that is so fashionable at the moment, and which is seen as 'revolutionary' is, as you point out, good old fashioned associative learning (albeit with a great deal of opacity in terms of working out what exactly has been learned).

Overall, and this is something that rarely happens, I can think of no suggestions for improvement, as the paper makes its case clearly, and well and will make a valuable and important contribution to the social learning literature, and to the field of animal cognition in general.

Decision letter (RSOS-181777.R0)

11-Feb-2019

Dear Dr Lind

On behalf of the Editors, I am pleased to inform you that your Manuscript RSOS-181777 entitled "Social learning through associative processes: A computational theory" has been accepted for publication in Royal Society Open Science subject to minor revision in accordance with the referee suggestions. Please find the referees' comments at the end of this email.

The reviewers and handling editors have recommended publication, but also suggest some minor revisions to your manuscript. Therefore, I invite you to respond to the comments and revise your manuscript.

- Ethics statement

- Data accessibility

If you wish to submit your supporting data or code to Dryad (<http://datadryad.org/>), or modify your current submission to dryad, please use the following link:
<http://datadryad.org/submit?journalID=RSOS&manu=RSOS-181777>

- Competing interests

- Authors' contributions

- Acknowledgements

- Funding statement

Because the schedule for publication is very tight, it is a condition of publication that you submit the revised version of your manuscript before 20-Feb-2019. Please note that the revision deadline will expire at 00.00am on this date. If you do not think you will be able to meet this date please let me know immediately.

on behalf of Dr Alecia Carter (Associate Editor) and Professor Kevin Padian (Subject Editor)
openscience@royalsociety.org

Associate Editor Comments to Author (Dr Alecia Carter):

Associate Editor: 1

Comments to the Author:

Dear Authors, I have now received three reviews of your manuscript and read it myself. All three reviewers were complimentary of the manuscript and could find little to comment on. I find myself in agreement with the reviewers – I found the manuscript interesting, well-written and timely, and found that it raises some interesting ideas for experiments.

Minor comments:

L361: learning, For -> learning. For

L400: truck Formally -> truck. Formally

L421: speedup -> speed-up

L423: might be better to change “even quicker” to “even more quickly”

L508: of behaviour -> on behaviour

It’s not important but although stated on L35 that the authors will use the term “animals” to refer to non-human animals, “non-human animals” is used (e.g. LL633, 709)

Reviewer comments to Author:

Reviewer: 1

Comments to the Author(s)

My thanks to the Editor and the Authors for the opportunity to review the present paper.

Social learning theory has grown dramatically in recent years. One focus has been its underlying cognitive mechanisms. The submitted paper explores how social learning phenomena can be explained by a combination of genetic predispositions and associative learning. The Authors introduce a simple model to formalise associative learning of (social, non-social, or a mix of both) stimuli that covers a surprising range of social learning terms. I find the paper compelling. Just as minimal neural networks give an idea of how complex a mechanism has to be to solve a given problem, the present model gives insight into the minimal requirements for solving a given social learning situation. Some specific comments follow, most of which are minor.

1. ll. 53–56 While I generally agree with the statement, it appears to be a bit too general. Copying the majority, for example, can be seen as social learning but does not require the majority to be experienced (e.g. mate-choice copying in *Drosophila*). Social learning can also be useful for updating information (e.g. waggle dance communication in honey bees).

2. l. 100 “... if a sequence of behavior is considered rather than a single a response.”: Is this sentence supposed to end with “behavior”?

3. l. 257 Doesn’t the stimulus-response association approach 25 in figure 1?

4. ll. 277–278 “Somewhat surprisingly, the model predicts that animals with smaller behavioral repertoires are more likely to learn to imitate.”: Where is this shown?

5. following l. 315 eq. (9) refers to the non-social stimulus as S_X , whereas in the text above and below it seems to be referred to as X only. Is there a difference between the two? Personally, S_X might be the better option for a non-social stimulus as it follows the nomenclature of S_{social} . Maybe it is worth changing it in the rest of the manuscript.

6. l. 371 “... sequences of l behaviours.”: Should this be “of length l ”?

7. l. 441 “sequence backwards by starting at S_n ” If the sequence started from the end should it then start with S_1 ?

8. I. 454 “simulations such as the one in 5 can be conducted.” Should that be a reference to section 3.5?

Reviewer: 2

Comments to the Author(s)
Review of RSOS-181777

This paper describes a model for associative learning taking into account various mechanisms and sequences of behaviours and stimuli from a social perspective. I think the writing was fantastic and have very few comments. My one confusion by the end of it was the applicability or usefulness of the model for those of us that study learning. The models were created and run, giving a nice overview of expectations I guess under certain learning scenarios. However I think that the purpose of the models were a little bit muddled by so much discussion. I would just suggest to make the aim of the model, and its potential applicability to future studies much more clear throughout the paper. In other words, why bother doing this, or why should others bother thinking about it? Can it help us just to understand? Or does it give us a really nice theoretical framework in which to interpret our own experimental results? (I would suggest both). I think just reinforcing the value of this computation model and its potential to help researchers analyse, interpret and discuss their own work in context would help the paper reach its full potential. Other than this, I think it's wonderful work and I really appreciate the effort that has gone into this!

Minor comments

Line 25 – I'm not sure what you mean by acquiring “productive behaviour” – is there another way to describe what you are trying to say? Can you clarify?

Line 26 – One of the major benefits of social learning is reducing individual risk (rather than, say, energy expenditure). I'd suggest mentioning this here.

Line 29 – You should probably provide a short definition of what you mean by traditions and/or culture here, considering you're just trying to introduce the whole topic your terms need to be very clearly defined.

Lines 64-65 – You say “as a benchmark” twice in the sentence.

Line 59 (and elsewhere) – Can you just define the aim of your model a little bit more? Your title implies it will be computational, will it therefore be predictive or simply explanatory? Do you hope it will be used by others as a framework in which to analyse their own data or experiments? What are you hoping is the outcome from introducing this model, how will this model serve the field specifically? Just a bit more here would be great.

Line 100 – Remove “a” before the word response.

Line 167 – This is great, a really important feature allowing for individual actions to reinforce (or not) particular learned behaviours.

Line 637 – I would say that there is an exception to this rule in birdsong: most studies have used production imitation as the explanatory process of learning to sing (either through active tutoring

or eavesdropping, or even copying non-biotic sounds somehow, such as in Superb lyrebirds or mockingbirds). It is thus widespread in passerines in this particular context (although they need a complex set of brain structures and a highly developed auditory and vocalization pathway for this) – the sequences are based on auditory memories (but take a lot of self-reinforcement). I would probably mention this because song is not mentioned much in this paper yet this is a major theme for some of us that study learning and cognition. It's an interesting one because while there is a reward, this reward only comes after a very long time, once the energy has been put in (for months or years sometimes) of imitation and production. The motivations are still very unclear, but thought to have a genetic component.

Reviewer: 3

Comments to the Author(s)

This is another in a great series of papers showing how certain behaviours that are thought to require specialized and/or complex cognitive mechanisms can be produced via associative learning. I really enjoyed reading the paper, and thought you did an excellent job of showing how your model could account for many different kinds of social learning in a clear and accessible way.

One (very) minor but valuable point that needs to be made over and over is that the "deep learning" that is so fashionable at the moment, and which is seen as 'revolutionary' is, as you point out, good old fashioned associative learning (albeit with a great deal of opacity in terms of working out what exactly has been learned).

Overall, and this is something that rarely happens, I can think of no suggestions for improvement, as the paper makes its case clearly, and well and will make a valuable and important contribution to the social learning literature, and to the field of animal cognition in general.

Author's Response to Decision Letter for (RSOS-181777.R0)

See Appendix A.

Decision letter (RSOS-181777.R1)

18-Feb-2019

Dear Dr Lind,

I am pleased to inform you that your manuscript entitled "Social learning through associative processes: A computational theory" is now accepted for publication in Royal Society Open Science.

You can expect to receive a proof of your article in the near future. Please contact the editorial office (openscience_proofs@royalsociety.org and openscience@royalsociety.org) to let us know if

you are likely to be away from e-mail contact. Due to rapid publication and an extremely tight schedule, if comments are not received, your paper may experience a delay in publication.

on behalf of Dr Alecia Carter (Associate Editor) and Kevin Padian (Subject Editor)
openscience@royalsociety.org

Appendix A

Reply to reviewers

Lind, Ghirlanda & Enquist

February 15, 2019

We would like to thank the reviewers and the editor for their comments, and for taking interest in our work. Please find all comments and our replies below (comments in italics and replies in upright font).

1 Associate editor comments

- *Dear Authors, I have now received three reviews of your manuscript and read it myself. All three reviewers were complimentary of the manuscript and could find little to comment on. I find myself in agreement with the reviewers—I found the manuscript interesting, wellwritten and timely, and found that it raises some interesting ideas for experiments.*

We are grateful for the positive responses to our manuscript. Please find below our response to all issues raised.

1.1 Minor comments

- *L361: learning, For -> learning. For*
- *L400: truck Formally -> truck. Formally*
- *L421: speedup -> speed-up*
- *L423: might be better to change “even quicker” to “even more quickly”*
- *L508: of behaviour -> on behaviour*
 - *It’s not important but although stated on L35 that the authors will use the term “animals” to refer to nonhuman animals, “nonhuman animals” is used (e.g. LL633, 709)*

Thanks for these corrections. They have all been changed in accordance with the suggestions, and the redundant use of “non-human” has been omitted.

2 Reviewer 1

2.1 Major points

- *Social learning theory has grown dramatically in recent years. One focus has been its underlying cognitive mechanisms. The submitted paper explores how social learning phenomena can be explained by a combination of genetic predispositions and associative learning. The Authors introduce a simple model to formalise associative learning of (social, nonsocial, or a mix of both) stimuli that covers a surprising range of social learning terms. I find the paper compelling. Just as minimal neural networks give an idea of how complex a mechanism has to be to solve a given problem, the present model gives insight into the minimal requirements for solving a given social learning situation. Some specific comments follow, most of which are minor.*

2.2 Detailed comments

1. *ll. 53–56 While I generally agree with the statement, it appears to be a bit too general. Copying the majority, for example, can be seen as social learning but does not require the majority to be experienced (e.g. matechoice copying in *Drosophila*). Social learning can also be useful for updating information (e.g. waggle dance communication in honey bees).*

To avoid misunderstandings we omitted “all”, and changed “logic” to “adaptive value”. The changed sentence now reads: “This diversity notwithstanding, the adaptive value of social learning is that inexperienced individuals can learn to behave more efficiently by using social information from experienced individuals. ”.

2. 1. 100 “... if a sequence of behavior is considered rather than a single a response.”: Is this sentence supposed to end with “behavior”?

No, here we have used “response” instead of “a behavior”. But, the sentence contained an extra “a” making it strange. The extra “a” has been omitted.

3. *l. 257 Doesn't the stimulusresponse association approach 25 in figure 1?*

This is correct, thank you for pointing this out. The text has been changed and “10” was replaced with “25”. We checked the scripts in supplementary information, and they were correct and consistent with text and figures in the ms.

4. *ll. 277–278 “Somewhat surprisingly, the model predicts that animals with smaller behavioral repertoires are more likely to learn to imitate.”: Where is this shown?*

This is a good point. We haven't elaborated on this, but it is a consequence of the model in that animals have repertoires of different sizes and that animals explore new situations probabilistically using their behavior repertoire. We elaborate on the combinatoric argument in Learning a behavior sequence (line 371). To make this point less obscure we have added the following sentence to this part: "In fact, all else being equal, the likelihood that an individual will learn to use the same behavior as another is greater in species with a smaller repertoire, simply because there are fewer behaviors to choose from (see also 3.4 for discussion on combinatorial matters)."

5. *following l. 315 eq. (9) refers to the nonsocial stimulus as S_X , whereas in the text above and below it seems to be referred to as X only. Is there a difference between the two? Personally, S_X might be the better option for a nonsocial stimulus as it follows the nomenclature of S_{social} . Maybe it is worth changing it in the rest of the manuscript.*

Thanks for pointing this out, we have changed throughout in accordance with the suggestion. Now non-social stimuli are always referred to as S_x (and e.g. S_y). We have also changed this in the figures.

6. *l. 371 "... sequences of l behaviours.": Should this be "of length l "?*

Yes, thanks for correcting this. We changed this so it now reads: "An animal with a repertoire of n behaviors, for example, can try out n^l behavior sequences of length l ."

7. *l. 441 "sequence backwards by starting at S_n " If the sequence started from the end should it then start with S_l ?*

Thanks again for finding this incorrect detail, we have changed to the correct S_l .

8. *l. 454 "simulations such as the one in 5 can be conducted." Should that be a reference to section 3.5?*

Thanks for pointing this out, we had missed the word "figure". It now reads "as the one in figure 5 can be conducted."

3 Reviewer 2

3.1 Major points

- *This paper describes a model for associative learning taking into account various mechanisms and sequences of behaviours and stimuli from a social perspective. I think the writing was fantastic and have very few comments. My one confusion by the end of it was the applicability or usefulness of the model for those of us that study learning. The models were created and*

run, giving a nice overview of expectations I guess under certain learning scenarios. However I think that the purpose of the models were a little bit muddled by so much discussion. I would just suggest to make the aim of the model, and its potential applicability to future studies much more clear throughout the paper. In other words, why bother doing this, or why should others bother thinking about it? Can it help us just to understand? Or does it give us a really nice theoretical framework in which to interpret our own experimental results? (I would suggest both). I think just reinforcing the value of this computation model and its potential to help reserchers analyse, interpret and discuss their own work in context would help the paper reach its full potential. Other than this, I think it's wonderful work and I really appreciate the effort that has gone into this!

We thank the reviewer for the comments. We agree that there is a lot of discussion of different aspects of social learning in our manuscript. Our attempt was to focus on a few cases that have some general applications, going from simpler to more complex cases. We are of course all for “*reinforcing the value of this computation model and its potential to help reserchers analyse, interpret and discuss their own work in context would help the paper reach its full potential.*”. In response to this request by reviewer 2, and to emphasize and clarify the potential applicability of our ms, we have added the following to the abstract. “Simulations were performed using a new learning simulator program. The simulator is publicly available and can be used for further theoretical investigations, and to guide empirical research of learning and behavior.” We hope these sentences together with the end of the abstract better showcase the scope and applicability of the ms.

3.2 Minor comments

1. *Line 25 – I’m not sure what you mean by acquiring “productive behaviour” – is there another way to describe what you are trying to say? Can you clarify?*

To avoid misunderstandings we have clarified this and changed this part to: “At the individual level, social learning helps naïve individuals acquire information from more experienced individuals resulting in behaviors that have positive outcomes or results in avoidance of negative ones. This saves time and energy and reduces individual risk, ultimately enhancing survival and reproduction.”

2. *Line 26 – One of the major benefits of social learning is reducing individual risk (rather than, say, energy expenditure). I’d suggest mentioning this here.*

We added reducing individual risk to this sentence. This part now reads: “thereby saving time and energy, and reducing individual risk, ultimately enhancing survival and reproduction”.

3. *Line 29 – You should probably provide a short definition of what you mean by traditions and/or culture here, considering you’re just trying to introduce the whole topic your terms need to be very clearly defined.*

We clarified this by adding a citation that delves into the details and definitions of traditions and cultures. In the text, after mentioning traditions, we added “(sensu Galef, 1992)”. This citation points to the paper “The question of animal culture” (Human Nature, 1992, 3(2), 157-178).

4. *Lines 6465 – You say “as a benchmark” twice in the sentence.*

Thanks for this correction.

5. *Line 59 (and elsewhere) – Can you just define the aim of your model a little bit more? Your title implies it will be computational, will it therefore be predictive or simply explanatory? Do you hope it will be used by others as a framework in which to analyse their own data or experiments? What are you hoping is the outcome from introducing this model, how will this model serve the field specifically? Just a bit more here would be great.*

As mentioned above, we expanded the abstract pointing out that we think our model can be used both for future theoretical research and as a guide for empirical tests of animal behavior. We do specify that our aim is to see if our model can account for social learning phenomena in animals (line 60-62). But, to better introduce our model and its scope, we added the following to the text: “This model, closely related to optimization algorithms in machine learning, can learn optimal behavior in ecologically relevant circumstances. This model has previously been shown capable of producing ‘intelligent’ behavior, such as tool use, self-control, and planning (Enquist et al. 2016, Lind 2018).”

6. *Line 100 – Remove “a” before the word response.*

The “a” was omitted, thanks.

7. *Line 167 – This is great, a really important feature allowing for individual actions to reinforce (or not) particular learned behaviours.*

Thank you!

8. *Line 637 – I would say that there is an exception to this rule in birdsong: most studies have used production imitation as the explanatory process of learning to sing (either through active tutoring or eavesdropping, or even copying nonbiotic sounds somehow, such as in Superb lyrebirds or mockingbirds). It is thus widespread in passerines in this particular context (although they need a complex set of brain structures and a highly developed auditory and vocalization pathway for this) – the sequences are based on auditory memories (but take a lot of selfreinforcement). I would probably mention this because song is not mentioned much in this paper yet this is a major theme for some of us that study learning and cognition. It’s an interesting one because while there is a reward, this reward only comes*

after a very long time, once the energy has been put in (for months or years sometimes) of imitation and production. The motivations are still very unclear, but thought to have a genetic component.

This is an interesting question raised by reviewer 2. Luckily, when we explored memory for stimulus sequences in Ghirlanda, Lind, & Enquist (2017, also in Royal Society Open Science) we also found data for songbirds (including starlings that are capable of imitating sounds). All studies available suggests that there seems to be no fundamental differences between how songbirds (capable of song learning and vocal imitation) recognize and remember sequences of arbitrary stimuli in comparison with other birds (e.g. pigeons) and mammals (e.g. rats, dogs, and macaques). This means that although imitation is common in some birds (such as starlings), it cannot develop through “production imitation” because production imitation requires faithful memory of stimulus sequences, which is something that appears to be absent in all nonhuman animals that so far have been tested.

However, we do agree that song learning is an important aspect of social learning, and importantly song learning contains important aspects of genetic specializations for social input. For this reason we end the ms with pointing out the importance of song learning as an example of social learning that is beyond the scope of our model. A part of the final paragraph reads:

“we concur with Heyes (2012) that “the suggestion that social learning is mediated by associative processes does not imply that all learning is associative.” In non-human animals, exceptions exist in systems with strong genetic support, such as in song learning in hummingbirds, passerines, and parrots (Catchpole and Slater 2003).”

Thanks also to reviewer 2 for pointing this out, because it lead us to correct the citation for bird song. It should of course be Catchpole and Slater 2003, not Janik and Slater 1997.

4 Reviewer 3

4.1 Major points

- *This is another in a great series of papers showing how certain behaviours that are thought to require specialized and/or complex cognitive mechanisms can be produced via associative learning. I really enjoyed reading the paper, and thought you did an excellent job of showing how your model could account for many different kinds of social learning in a clear and accessible way. One (very) minor but valuable point that needs to be made over and over is that the "deep learning" that is so fashionable at the moment, and which is seen as 'revolutionary' is, as you point out, good old fashioned associative learning (albeit with a great deal of opacity in terms*

of working out what exactly has been learned). Overall, and this is something that rarely happens, I can think of no suggestions for improvement, as the paper makes its case clearly, and well and will make a valuable and important contribution to the social learning literature, and to the field of animal cognition in general.

We would like to thank reviewer 3 for these kind comments. In response to the “deep learning” comment, we changed a sentence in the final paragraph of the Introduction to connect with this literature. This sentence now reads: “This model, closely related to optimization algorithms in machine learning that are commonly used in for example “deep learning” studies (Goodfellow et al. 2016), can learn optimal behavior in ecologically relevant circumstances.”